# Enhancing Passion Fruit Resilience: The Role of Hariman in Mitigating Viral Damage and Boosting Productivity in Organic Farming Systems

**DOI:** 10.3390/ijms26052177

**Published:** 2025-02-28

**Authors:** José Leonardo Santos-Jiménez, Caroline de Barros Montebianco, Mariana Collodetti Bernardino, Eliana Barreto-Bergter, Raul Castro Carriello Rosa, Maite Freitas Silva Vaslin

**Affiliations:** 1Plant Molecular Virology Laboratory, Departamento de Virologia, Instituto de Microbiologia, Universidade Federal do Rio de Janeiro (UFRJ), Av. Carlos Chagas, Filho, 373, CCS, Rio de Janeiro 21941-599, RJ, Brazil; jose@micro.ufrj.br (J.L.S.-J.); cbmontebianco@micro.ufrj.br (C.d.B.M.); mcollodetti@gmail.com (M.C.B.); 2Tolveg Inc., Incubadora de Empresas COPPE UFRJ, Rua Hélio De Almeida s/n—Prédio, 1 Sala 10, Ilha do, Fundão, Rio de Janeiro 21941-972, RJ, Brazil; 3Laboratory of Biological Chemistry for Microorganisms, Instituto de Microbiologia, Universidade Federal do Rio de Janeiro (UFRJ), Av. Carlos Chagas Filho, 373, CCS, Rio de Janeiro 21941-599, RJ, Brazil; eliana.bergter@micro.ufrj.br; 4Embrapa Agrobiologia, Empresa Brasileira de Pesquisa Agropecuária (EMBRAPA), Seropédica 23897-970, RJ, Brazil; raul.rosa@embrapa.br

**Keywords:** biostimulants, bioinputs, CABMV, viral tolerance, defense genes induced after elicitation, phytohormone elicitation, organic farming, sustainable crop productivity improvement, plant growth promotion

## Abstract

This study investigates the molecular mechanisms by which Hariman mitigates damage and productivity losses caused by Cucumber Aphid-Borne Mosaic Virus (CABMV) in the passion fruit genotypes ‘FB300’ and ‘H09-110/111’ under greenhouse and field conditions in Rio de Janeiro, Brazil. Hariman treatment induced the upregulation of key defense genes and phytohormones in response to CABMV infection, enabling treated plants to counteract virus-induced developmental impairments effectively. The relative accumulation of CABMV and disease severity were significantly reduced, with treated plants showing no decline in growth parameters such as height, leaf count, flower production, or fruit set. Over 18 months, total productivity increased by 65.7% and 114% for ‘FB300’ and by 44% and 80% for ‘H09-110/111’ after one and two applications of Hariman, respectively. Notably, infected plants treated with Hariman outperformed healthy plants grown under similar conditions, underscoring the biofertilizer’s dual role in promoting plant growth while enhancing resistance to biotic stressors. These findings indicate that Hariman stimulates robust growth and induces the expression of the defense-related genes *PR-3*, *SOD*, *POD12*, *PAL*, and *LOX2* alongside the expression of the phytohormone-associated genes *SAUR20* and *GA2ox* across different passion fruit genotypes. The adoption of these sustainable technologies holds significant potential for enhancing passion fruit productivity in the face of diseases that severely threaten this crop.

## 1. Introduction

The genus *Passiflora* L. belongs to the Passifloraceae family, comprising about 500 species, among which, *Passiflora edulis* stands out for its economic and medicinal importance [1]. It is widely planted in tropical and subtropical regions around the world, especially in South America, the Caribbean, Florida, South Africa, and Asia [1]. With approximately 200 species, Brazil is a center of origin of passion fruit species [2]. Among the different genotypes of passion fruit, the yellow-fruited *P. edulis* f. flavicarpa and the purple-fruited type, *P. edulis* Sims, are the cultivars that have the greatest economic and commercial importance [3]. *P. edulis* f. flavicarpa develops fruits with a bright yellow peel and a hard, thick consistency, measuring approximately 6 to 12 cm in length and 4 to 7 cm in diameter. They have an acidic pulp with a strong aromatic flavor and brown seeds. The purple fruits from *P. edulis* Sims have a purple peel; are smaller in size, measuring approximately 4 to 9 cm in length and 3.5 to 7 cm in diameter; and have black seeds [4].

Brazil is recognized as the world’s largest producer of passion fruit, with an impressive annual output of 697,859 tons, representing approximately 70% of global production and an average national yield of 15.3 tons per hectare [5]. In organic production systems, however, there is significant variability in productivity, which ranges from 7.5 to 18.8 tons per hectare [6]. The primary species cultivated for commercial purposes is *P. edulis*, which is known to be susceptible to various viral infections that have been documented globally [7]. In Brazil, the cowpea aphid-borne mosaic virus (CABMV) emerges as the most critical and widespread viral pathogen, raising serious concerns across all cultivation regions [8]. Additionally, several other viruses have been identified, with begomoviruses presenting a particularly pressing challenge [9,10,11,12]. This growing body of evidence underscores the need for ongoing research and strategic interventions to mitigate the impact of viral infections on passion fruit production.

CABMV, belonging to the genus *Potyvirus* in the family Potyviridae, has a positive-sense, single-stranded RNA molecule approximately 10 kilobases in length encased in flexible and filamentous particles. CABMV is transmitted and disseminated mechanically during pruning and by some species of aphids (*Aphididae* family, *Hemiptera* order) in a non-persistent manner. Although there are no aphid species colonizing *Passiflora* spp., a few species have been described as CABMV vectors, specifically *Aphis gossypii* Glover, *A. craccivora* Bock, *A. fabae* Scopoli, *Myzus persicae* Shulzer, *Uroleucon ambrosiae* Thomas, *U. sonchi* L., *Toxoptera citricidus* Kilkaldy, and *Myzus nicotianae* Blackman [13,14].

Passion fruit woodiness disease (PWD), caused by CABMV, causes serious damage to passion fruits. The disease causes severe symptoms, including fruit hardening and distortion, which reduce crop quality and yield. The infection decreases fruit size and causes deformations associated with the occurrence of mosaic symptoms, accompanied by leaf malformation, blisters, a marked reduction in plant development, and woodiness in some fruits [15]. Recent studies have identified various wild Passiflora genotypes with resistance to CABMV, with some potentially immune species and others simply being asymptomatic carriers [8]. However, there have been no resistant commercial genotypes available until now.

Plants have developed systems to fight infections, activating specific defensive pathways aimed at various types of threats. The handling of plants with biotic elicitors is a useful approach to enhance the production or accumulation of important secondary metabolites [16]. The application of elicitors to plants can play various roles in improving, for example, the synthesis of important secondary metabolites while the treatments alter the plant’s physiological conditions. These treatments protect the plants against various biotic and abiotic stresses, such as attacks by pathogens, wounds, heavy metals, air pollutants, ultraviolet rays, and adverse environmental conditions [17].

Following infection by phytopathogens, plants activate a range of proteins, particularly phytohormones and signaling molecules, which are integral to their immune response and defense mechanisms. Key compounds such as salicylic acid (SA) and jasmonic acid (JA) trigger the accumulation of defense proteins, collectively known as pathogenesis-related proteins (PRs). While some PR proteins are present at low levels in healthy tissues, their concentrations increase rapidly upon pathogen exposure, serving as an early defense mechanism. They utilize enzymes that are essential for protection mechanisms from phytopathogens such as β-1,3-glucanase, chitinase, peroxidase, and ribonucleases [18,19]. Currently, 17 families of PR proteins have been classified, many of which are associated with specific biochemical functions, providing antibacterial or antifungal activities. In addition to their critical role in plant defense, phytohormones are also vital for nearly all aspects of plant development and growth processes [20,21]. This multifaceted role underscores the importance of understanding the complex interactions between plant immune responses and developmental pathways, which may enhance our capacity to improve plant resilience against biotic stressors.

Our group recently demonstrated that a *Cladosporium herbarum* (Davidiellaceae family, Capnodiales Order, *Fungi*)-derived peptidogalactomannan, commercial name Hariman, induced defense-related genes in passion fruits (*Passiflora edulis* Sims f. flavicarpa Deg) under greenhouse conditions that enable the plant to mitigate the symptoms of CABMV infection [22]. Here, Hariman elicitation was evaluated under field conditions, with plants submitted to multiple adverse conditions such as temperature variation, water stress, and pathogen attack. Our results showed that Hariman pre-treatment induced defense and phytohormone-associated gene induction, protecting our passion fruit plants of various genotypes against developing CABMV symptoms while inoculated under field conditions. This response led to important yield and productivity improvements in our organic production system.

## 2. Results

### 2.1. Hariman Treatment Is Effective in the Decrease of CABMV Disease Incidence and Severity in Field

In a previous study, we demonstrated that Hariman treatment significantly reduced both the incidence and severity of disease in ‘Redondo Amarelo’ passion fruits infected with cowpea aphid borne mosaic virus (CABMV) [22]. To further investigate the efficacy of Hariman treatment, we evaluated its effects on different passion fruit genotypes, specifically ‘FB300’ and ‘H09-110/111’, under various growing conditions. To evaluate the severity of the infection, characteristic leaf symptoms (including mosaic patterns, deformation, and blistering) were assessed in these genotypes following treatment with Hariman and subsequent CABMV application.

The incidence (DI) and severity (DS) of passion fruit woodiness disease were measured under both greenhouse and field conditions. In the greenhouse setting, we observed statistically significant differences (Dunn’s test, *p* < 0.0001) in DI and DS between the plants treated with Hariman + CABMV and those treated with water + CABMV across all the time points evaluated (Appendix A). Notably, the DI consistently remained lower in Hariman + CABMV plants over time. Additionally, at 9 wai, a reduction of 47% in DS was recorded for ‘FB300’ and 32% for ‘H09-110/111’ in plants treated with Hariman compared with the water control (Appendix A).

In field trials, infected plants treated with Hariman also exhibited lower DI and DS compared with those receiving water treatment. Significant differences were noted at 4 wai for both ‘FB300’ and ‘H09-110/111’ genotypes (Table 1). Interestingly, no significant differences in DI or DS were detected when comparing one versus two applications of Hariman treatment. These findings suggest that Hariman treatment is effective in managing CABMV infection across various passion fruit genotypes and growing conditions, warranting further exploration of its application in broader agricultural practices.

### 2.2. Analysis of Viral Accumulation in Passion Fruit Plants Treated with Hariman

The viral accumulation was analyzed in young leaves of plants across all treatments using qualitative RT-PCR, semi-quantitative ELISA, and quantitative RT-qPCR assays. Under field conditions, ELISA assays did not detect the presence of CABMV at 4 wai in plants treated with Hariman + CABMV for both ‘FB300’ and ‘H09-110/111’. In contrast, the virus was detected in plants from ‘FB300’ treated with water + CABMV (Figure 1a,b; Appendix A). The same was observed by using RT-PCR for CABMV diagnosis, except for ’FB300’, where very faint amplification bands were observed in the first three plants, indicating the presence of the virus in those plants. However, this was at much lower concentrations than in water + CABMV (Figure 1c). By 8 wai, all evaluated plants infected with CABMV, including ‘FB300’ and ‘H09-110/111’, exhibited detectable levels of the virus (Figure 1d). Notably, the genotype ‘H09-110/111’ demonstrated a significant reduction in CABMV levels as measured by ELISA when treated with Hariman compared with the water treatment (Figure 1b; Appendix A); only six out of twelve Hariman-treated plants tested positive for the virus via RT-PCR (Figure 1d). At 12 wai, a decrease in CABMV accumulation was observed in both genotypes that received one or two applications of Hariman, showing significant reductions of 28.9% and 22.7%, respectively, compared with the water + CABMV control (Figure 1a; Appendix A) (*p* < 0.001). The levels of the virus remained 18% and 14% lower in plants treated with one and two applications of Hariman, respectively, at this time point for plants of ‘H09-110/111’ (Figure 1b; Appendix A) (*p* < 0.001). Quantitative RT-qPCR results corroborated these findings, indicating a decrease in relative CABMV accumulation in Hariman + CABMV plants under both greenhouse and field conditions compared with those receiving water + CABMV. In greenhouse trials, CABMV was undetectable in any of the treated ‘Redondo Amarelo’ plants at 12 hai, while a significant decrease in Hariman + CABMV levels was noted at 168 hai (Figure 1e) (*p* < 0.05). Field experiments conducted approximately three months post-CABMV inoculation revealed that relative CABMV accumulation remained higher in water-treated plants than in those treated with Hariman, with statistically significant differences emerging at 12 wai for both ‘FB300’ (Figure 1f) and ‘H09-110/111’ (Figure 1g) (*p* < 0.05). Notably, virus coat protein transcripts were not detected in ‘H09-110/111’ plants treated with Hariman. These results suggest that Hariman treatment enhances the plant’s ability to withstand viral infections by reducing virus replication and limiting translocation to younger leaves.

### 2.3. Biostimulant Hariman Mitigates Viral Damage and Boosted Productivity

Among the detrimental effects caused by CABMV infection in passion fruit plants, a significant plant developmental delay was observed, leading to productivity losses in affected fields [22]. Our research group previously demonstrated that treatment with the biostimulant Hariman can alleviate developmental impairments caused by CABMV under greenhouse conditions. Consequently, we aimed to evaluate the efficacy of Hariman on two genotypes of *Passiflora edulis* in field conditions. Thirty-day-old ‘FB300’ and ‘H09-110/11’ plantlets were subjected to a single spray of 100 μg·mL^−1^ of Hariman or water (control treatment). Three days post-treatment, all plants were inoculated with CABMV. Untreated, uninoculated plants served as healthy controls. A portion of the plantlets were transferred to field conditions, while the remainder continued to grow in a greenhouse.

Seven weeks after CABMV inoculation (wai), Hariman-treated ‘H09-110/111’ plants exhibited development patterns comparable to or exceeding those of uninoculated plants, particularly in root formation under greenhouse conditions (Figure 2a). Furthermore, these plants demonstrated a biomass (both fresh and dry weights) similar to that of the healthy controls. In contrast, water-treated plants infected with CABMV experienced significant weight reductions attributable to CABMV infection. Specifically, the fresh and dry weights of the roots from Hariman-treated CABMV-infected plants were 100% and 71% greater than those of water-treated CABMV-infected plants, respectively (Figure 2b,c) (*p* < 0.001). The aerial parts of the Hariman-treated infected plants also exhibited increases of 86% and 33% in fresh and dry weights when compared with water + CABMV plants (Figure 2d,e) (*p* < 0.01).

Additionally, significant increases in plant height and leaf count were observed for Hariman + CABMV-treated plants compared with water + CABMV treatments (*p* < 0.05). Specifically, height increased by 26% for ‘FB300’ and 17% for ‘H09-110/11’ at 5 wai (Figure 2f,g) (*p* < 0.001), while both genotypes showed over a 30% increase in leaf number at 9 wai (Figure 2h,i) (*p* < 0.001).

Data collected from plants grown under field conditions revealed significant enhancements in leaf count (40%) and height (30%) for Hariman + CABMV-treated plants from both ‘FB300’ and ‘H09-110/111’ compared with water + CABMV-treated counterparts (Figure 3a,b,d,e) (*p* < 0.001). A subset of these field-planted plants received a second application of Hariman at 8.5 wai; however, no significant differences were noted between single or double applications regarding height or leaf number for either variety.

A notable effect of Hariman treatment was observed in earlier flower induction across both treatment groups. Some treated passion fruit plants began forming flower buds at 10 wai. Although no differences were recorded between single or duplicate applications regarding initial flower bud counts, significant differences emerged at 12 wai, with an approximate increase of 133% in flower bud numbers for both genotypes at 16 wai compared with water-treated controls (Figure 3c,f) (*p* < 0.001).

The overall appearance of both genotypes (‘FB300’ and ‘H09-110/111’) treated with Hariman, regardless of application frequency, was markedly superior throughout the experimental period, characterized by enhanced foliar development compared with control groups (Figure 3g–j,o–r and Appendix A). These empirical observations were corroborated by quantitative data reflecting increases in height, leaf count, and flower bud numbers, indicating that Hariman treatment effectively mitigates developmental symptoms associated with viral infection while also acting as a bioinductor.

As anticipated, total fruit number (TNF) also significantly increased in Hariman + CABMV-treated plants, both in those receiving one or two sprays, relative to controls (uninoculated and water + CABMV treatments) across both genotypes (‘FB300’ and ‘H09-110/11’, Table 2). Notably, plants receiving two applications of Hariman yielded a higher mean fruit count than other treatments (Table 2) (*p* < 0.001). The ‘FB300’ genotype showed the best responses, with 56.4% and 97.6% growth in TNF in single and duplicate Hariman treatments, respectively.

In addition to TNF, all fruit parameters demonstrated optimal results following one or two applications of Hariman (Figure 3k–n,s–v and Appendix A), including greater fresh weight (WF), pulp weight (WP), and pulp yield percentage (PY%) (Table 2) (*p* < 0.001).

Viral infections severely impact passion fruit productivity; however, over a single growing season lasting ten months, we observed that Hariman treatment effectively mitigated productivity losses associated with viral infections. Plants receiving two applications yielded approximately 15.5 tons per hectare, while those with a single application yielded about 12 tons per hectare across both genotypes. In contrast, control plants yielded approximately 7–8 tons per hectare (Table 2) (*p* < 0.0001). Thus, Hariman treatment resulted in productivity increases of approximately 65.75% and 114.31% for ‘FB300’ and ‘H09-110/11’, respectively, corresponding to increases of 44.48% and 80.02% for one versus two sprays.

Finally, an important quality parameter for passion fruit cultivation is sugar content; soluble solids (SS) were evaluated across all treatments with no significant differences detected among them (Table 2) (*p* > 0.05), suggesting that Hariman treatment does not alter sugar content in passion fruit pulp.

### 2.4. Impact of CABMV Infection on Defense-Related Gene Expression and Phytohormone Precursors

In our previous report, we observed that CABMV infection significantly affects the expression of several defense-related genes within the initial hours post-infection. This study aims to further investigate the gene expression profile and the levels of two phytohormone precursors until 12 wai in untreated plants.

The ‘Redondo Amarelo’ passion fruit plants maintained under greenhouse conditions were analyzed shortly after infection, revealing a suppression of three key defense-related genes: *PR-3*, *PAL*, and *LOX2*, as well as a downregulation of phytohormone-associated genes *SAUR20* and *GA2ox* at 12 hai (Figure 4a). Although CABMV was not detectable by RT-qPCR at this early time point (Figure 1c), our findings indicate that the virus rapidly induces molecular disruptions within the plant. Notably, the suppression of *PR-3*, *PAL*, and *GA2ox* persisted up to 168 hai (Figure 4a). Conversely, *SOD* was induced at both time points, suggesting it has a critical role in enhancing plant survival under stress conditions.

Under field conditions, ‘FB300’ CABMV-infected plants exhibited a downregulation of all defense-related genes analyzed, along with the two phytohormone precursors (Figure 4b). In contrast, CABMV-inoculated plants of the ‘H09-110/111’ genotype only displayed a downregulation of *SOD*, *SAUR20*, and *GA2ox* at this time point. Interestingly, *PR-3* and *POX12* were induced in this genotype, indicating a potentially more robust immune response compared with ‘FB300’ against CABMV infection (Figure 4b).

Both passion fruit commercial genotypes ‘Redondo Amarelo’ and ‘FB300’ are susceptible to CABMV infection. The observed gene modulation suggests that the virus induces an inhibition of critical defense mechanisms and phytohormone pathways, which may exacerbate the severity of CABMV infection.

### 2.5. Differential Expression of Defense-Related Genes in Passion Fruit Genotypes Under Hariman Treatment

The expression profiles of five defense-related genes were evaluated at 24 and 72 h after treatment (hat) in Hariman-treated passion fruit plants, specifically genotypes ‘FB300’ and ‘H09-110/111’, under greenhouse conditions. At 24 hat, ‘FB300’ only exhibited induction of the *POD12* and *LOX2* mRNAs, while all the other genes remained at baseline levels (Figure 5). However, a strong induction of all profiles was observed at 72 hat. Notably, the expression of the *LOX2* gene was significantly elevated, with levels approximately 120 times higher at 24 hat and around 40 times higher at 72 hat compared with water-treated baseline levels (Figure 5c). In contrast, the ‘H09-110/111’ genotype demonstrated a rapid induction of defense mechanisms. All evaluated genes were activated within 24 hat and returned to baseline levels at 72 hat, with the exception of the gene *PAL*.

For ‘FB300’ plants cultivated under field conditions, all analyzed genes were overexpressed at 8 wai but reverted to basal levels after four weeks, with the exception of *SOD*, which remained induced (Figure 5). Interestingly, plants treated with one or two Hariman applications exhibited similar expression levels; however, *PAL* was induced by a factor of 5.5 times and *LOX2* by a factor of 1.8 times more in plants receiving a single Hariman treatment compared with those receiving two treatments (Figure 5c,e). The results for ‘H09-110/111’ showed that, at 8 and 12 wai, *PAL* and *LOX2* remained strongly induced compared with infected control (water-treated) plants. Conversely, genes *PR3*, *SOD*, and *POD12* were not induced at 8 wai but exhibited significant induction four weeks later (Figure 5a,b,d). The frequency of Hariman sprays also influenced gene expression profiles; curiously, when plants received two sprays, all genes returned to basal expression levels.

Overall, these findings highlight the differential responses of passion fruit genotypes ‘FB300’ and ‘H09-110/111’ to Hariman treatment, emphasizing the importance of timing and treatment frequency in modulating defense-related gene expression. Furthermore, Hariman treatment appeared to enhance the expression of all defense-related genes that were previously suppressed or minimally induced following CABMV infection in both genotypes.

### 2.6. Impact of Hariman Treatment on Phytohormone Signaling in CABMV-Infected Plants

As previously discussed, CABMV infection is known to suppress the expression of genes associated with phytohormone signaling pathways. Our observations indicate that treatment with Hariman significantly induced the expression of *SAUR20* and *GA2ox* mRNAs. This induction was particularly pronounced in the initial hours following CABMV inoculation, specifically at 12 and 168 hai under greenhouse conditions (Figure 5f).

In field trials, plants of the ‘FB300’ and ‘H09-110/111’ cultivars infected with CABMV exhibited notable increases in the expression of these phytohormone-related genes at 12 wai with little difference between plants that received one or two applications of Hariman treatment (Figure 5g).

Consistent with our findings related to defense-related genes, Hariman treatment effectively mitigated the suppression of phytohormone-associated genes, particularly those linked to auxin and gibberellin pathways, which is typically induced by viral infection. This mechanism may partially explain the reduced damage observed in CABMV-infected plants following Hariman treatment.

### 2.7. Use of Hariman Alone and/or in Combination with Inoculants in a Commercial Farm Improved Productivity

Field studies conducted in 2023 in Brumado municipality, Bahia State, Brazil, evaluated the effects of Hariman treatment and in combination with Plant Growth Promoting Bacteria (PGPB) on the productivity of passion fruit. Treatments were applied in June, with yield analyses conducted five months later in October. The results highlighted substantial improvements in key productivity parameters: the total number of fruits (TNF), total yield (TY), fruit weight (WF), and pulp yield (PY). The application of twice administered Hariman treatment alone resulted in a remarkable 35.68% increase in total yield compared with the water-treated control group (Appendix A). In contrast, the combination of Hariman with PGPB yielded an impressive 62.46% increase in total yield when applied in two separate applications. These findings indicate that both treatments significantly enhance overall productivity in passion fruit cultivation. Furthermore, the evaluation of fruit weight revealed that the PGPB treatment notably contributed to an increased fruit size. However, it is important to note that the pulp yield did not exhibit significant differences among the various treatments, indicating that while the interventions improved total yield and fruit weight, they did not adversely affect the quality of the fruit pulp.

Overall, these results underscore the potential use of Hariman alone or together with PGPB in passion fruit production.

## 3. Discussion

Viral diseases present a significant challenge to passion fruit cultivation, necessitating effective protective treatments. This study evaluated the efficacy of the organic biostimulant peptidogalactomannan Hariman in mitigating infections caused by the CABMV virus in passion fruit plants. Our results demonstrate that treatment with Hariman significantly reduces both the incidence and severity of CABMV infections in greenhouse and field settings. Specifically, Hariman application not only alleviates disease symptoms but also markedly decreases viral accumulation in treated plants compared with untreated controls.

The observed reduction in virus replication and spread is likely a contributing factor to the decreased disease severity and symptom expression. Our findings are consistent with the existing literature that underscores the role of biostimulants in enhancing plant resilience against viral pathogens [23,24,25,26]. Notably, plants treated with Hariman prior to CABMV exposure exhibited enhanced defense mechanisms, indicated by the sustained expression of defense-related genes. This suggests that Hariman plays a crucial role in modulating plant responses to viral stress, thereby providing a protective effect against CABMV infection.

To elucidate the underlying mechanisms, we investigated the impact of CABMV infection on defense-related pathways and phytohormonal signaling. Our analysis revealed that CABMV infection suppresses key genes associated with plant defense and phytohormone pathways. Infected plants exhibited compromised root systems and overall growth reduction. Conversely, treatment with Hariman resulted in significant improvements in morphological traits and productivity parameters, highlighting the critical role of phytohormones in processes such as cell expansion, floral modulation, and fruit development [27,28,29].

These observations align with previous studies on virus–plant interactions. For instance, viral infections in passion fruit plants have been shown to cause significant changes in morphoagronomic characteristics, such as deformation, mottling, and reduced fruit quality. In addition, these viral infections cause changes at the molecular and biochemical level, such as protein synthesis related to signaling, metabolism, and defense mechanisms [7,30]. Similar patterns have been documented in other pathosystems [31,32,33].

In our study, pre-treatment with Hariman effectively prevented the downregulation of five critical defense-related genes at both 8 and 12 weeks after inoculation with CABMV. This strong induction of defense-related genes before and after CABMV inoculation likely contributes to the observed reduction in viral symptoms and disease severity. Furthermore, Hariman treatment appears to inhibit the downregulation of essential phytohormonal signaling pathways associated with CABMV infection, thereby mitigating adverse effects on plant development and enhancing productivity characteristics.

Our findings provide compelling evidence that peptidogalactomannan Hariman serves as an effective biostimulant for protecting passion fruit plants against CABMV infections. By enhancing plant defense mechanisms and reducing viral replication, Hariman not only diminishes disease severity but also promotes overall plant health and productivity. Further research is warranted to explore the broader implications of biostimulant applications in managing viral diseases across various crops.

Pathogenesis-related proteins (PRs) are recognized as key components in plant defense mechanisms against pathogens. In our study, we observed the significant upregulation of *PR-3* (chitinase I) and *PR-9* (peroxidase 12—*POD12*) within hours following Hariman treatment across various passion fruit genotypes. Previous research by Parkinson et al. demonstrated that acibenzolar-S-methyl (Bion^®^, Syngenta, Basel, Switzerland) enhances PR activity in passion fruit infected with Passion Fruit Woodiness virus, resulting in reduced disease severity [34]. Our findings corroborate these results, indicating that Hariman treatment induces *PR-3* and other defensive genes before and after CABMV inoculation while leading to a marked decrease in disease severity over time.

Notably, under field conditions, no viral presence was detected in Hariman-treated plants infected with CABMV at four weeks after inoculation. At later time points, we observed a decline in relative CABMV accumulation in Hariman + CABMV passion fruit plants, suggesting that Hariman may impede translocation and replication of CABMV within young leaves. This indicates a potential “turning off” of symptom manifestation, with induced defense-related genes contributing to enhanced tolerance against CABMV infection.

Our results indicate that Hariman treatment significantly elevates mRNA transcripts of reactive oxygen species-eliminating enzymes such as *POD12* and *SOD*, reinforcing the role of antioxidant metabolism in the response of passion fruit to CABMV infection. The involvement of salicylic acid (SA) as a critical signaling molecule against viral pathogens has been well-documented; increased levels of *PAL* mRNAs post-Hariman treatment suggest an induction of defense responses through isoflavonoid synthesis or SA production.

Plant–virus interactions are known to impede growth by altering phytohormone levels and their signaling pathways rapidly. Emerging perspectives suggest that viral interference can weaken defense strategies by enhancing viral replication and systemic spread. Our observations revealed the suppression of *SAUR20* and *GA2ox* at various time points under greenhouse conditions, consistent with previous reports on gene suppression due to viral infection [30].

Importantly, our data indicate that Hariman treatment prepares plants such that subsequent CABMV inoculation does not inhibit phytohormone-associated genes like *SAUR20* and *GA2ox*. Consequently, these plants exhibited normal development without severe damage to their root systems or aerial parts, leading to increased flowering and fruiting rates, which ultimately enhanced crop yields.

To better understand the mechanisms underlying the effects of Hariman treatment, we further explored the correlation between gene expression and virus accumulation in passion fruit plants. Our results suggest that Hariman treatment modulates key defense-related genes, including those involved in the salicylic acid (SA) and jasmonic acid (JA) signaling pathways. These phytohormones play crucial roles in plant defense, with SA primarily involved in activating defense responses against biotrophic pathogens and JA regulating responses to necrotrophic pathogens and herbivory [35,36,37]. It is well-documented that SA and JA interact to orchestrate a coordinated defense response by inducing the expression of defense genes and other defense mechanisms, which enhance the antiviral ability of plants [38,39,40,41]. In our study, Hariman treatment enhanced the expression of genes involved in both SA and JA signaling, suggesting that this biostimulant activates a broad defense network capable of limiting CABMV accumulation and mitigating disease severity.

Furthermore, our study evaluated the effects of Hariman treatment in both short- and long-term scenarios. In the short term, we assessed defense gene expression, phytohormone-related genes, morphoagronomic parameters, and virus accumulation under greenhouse and field conditions. In the long term, we extended our evaluations to include defense gene expression, phytohormone-related genes, and productivity parameters such as fruit yield and quality. Fruit yields were evaluated 10 months after Hariman treatment. These evaluations were conducted across three different locations: Seropédica and Rio de Janeiro (State of Rio de Janeiro) with semi-humid tropical climates, and Brumado (State of Bahia) with a semi-arid climate. Additionally, Santos-Jiménez et al. [42] evaluated the effect of Hariman in the municipality of Campos dos Goytacazes (State of Rio de Janeiro), at a 335 km distance from Seropédica. All of these studies observed a positive effect on the productivity parameters of Hariman-treated plants.

However, while Hariman’s effectiveness was evident in both short- and long-term evaluations, further research is needed to assess the durability of its protective effects over multiple growing seasons and under varying environmental conditions, such as different climate zones or soil types. Studies on the persistence of biostimulant effects in diverse settings are critical for understanding the practical viability of Hariman in real-world agricultural scenarios.

Although variability was observed in developmental parameters such as the number of leaves, plant height, and flower buds, this is common in field experiments, where factors such as individual plant differences, environmental conditions, and measurement methods can influence data dispersion. To minimize these effects, we implemented several measures, including the use of independent biological replicates (*n* = 12), standardized measurement methods, and a randomized block design. Additionally, rigorous statistical analysis was performed to ensure the robustness of the results. While variability is inherent in field studies due to uncontrollable factors, we believe our strategies were effective in mitigating its impact, thus supporting the reliability of our findings.

In this study, we achieved improved fruit quality and yield increases for both evaluated genotypes compared with water + CABMV treatments. Notably, our results gave an average yield surpassing national averages for yellow passion fruit under organic production systems. Treatments involving Hariman alone or combined with PGPB significantly enhanced total fruit numbers compared with controls.

## 4. Materials and Methods

### 4.1. Location of Experimental Areas

Experiments were carried out under greenhouse conditions, at the Federal University Rio de Janeiro (UFRJ), Rio de Janeiro, at SIPA (Integrated system of agroecological production), “Fazendinha Agroecológica”, which is an experimental area of UFRRJ, PESAGRO, and Empresa Brasileira de Pesquisa Agropecuaria (Embrapa Agrobiologia). It is located in the municipality of Seropédica, State of Rio de Janeiro, Brazil, at the geographical coordinates 22°48′00″ S latitude and 43°41′00″ W longitude. It has an altitude of approximately 33 m, and the soil is classified as red–yellow Argisol. According to the Köppen classification, the climate is Aw (rainfall concentrated from November to March, with an average annual rainfall of 1213 mm and an average annual temperature of 24.5 °C).

### 4.2. Plant Material

Experiments were carried out under greenhouse and field conditions. Various *Passiflora edulis* genotypes were used in this study: two commercial genotypes of yellow passion fruit such as ‘Redondo Amarelo’ and ‘FB300’, and an intraspecific hybrid from the Passion fruit Germplasm Active Bank of Embrapa, ‘H09-100/111’. All the experiments were performed in an organic production system according to the recommendations for yellow passion fruit cultivation in Brazil and Brazil’s federal laws of organic agriculture [43]. Under field conditions, control of spontaneous vegetation was carried out by means of mechanized weeding between rows and hoeing in rows when needed and pollination was naturally performed by bumblebees (*Xylocopa* spp.). No insecticides, fungicides, or chemical fertilizers were used during all experiments’ periods.

### 4.3. Hariman Synthesis and Treatment

*Cladosporium herbarum* fungus strain CBS 121,621 was grown in potato dextrose broth medium (PDB) for 7 days and 3MM paper filtered to obtain the fungal mass. The fungal mass was processed according to Haido et al. [44].

In all experiments, passion fruit plants were foliar sprayed with 100 μg.mL^−1^ of Hariman resuspended in tap water when our plants were at the stage of 3 to 4 true leaves. This dose was selected based on Santos-Jiménez et al. [22] and Mattos et al. [45]. Hariman treatment was applied once or twice in the field. A 60-day interval between applications was applied on experiments realized on Seropédica while an interval of 12 weeks between applications was implemented for field tests performed in Bahia. Hariman was sprayed using a coastal manual sprayer (Jacto—XP, Jacto, Pompéia, São Paulo, Brazil). As a control, tap water was foliar sprayed using the same procedure (referred as water treatment). Uninoculated plants were used as healthy controls.

### 4.4. CABMV Mechanical Inoculation

Three days after Hariman treatment, 100 µL of a CABMV 1:50 dilution inoculum was mechanically inoculated into the first two leaves of each plant using Celite (Sigma-Aldrich, St. Louis, Missouri, USA) as an abrasive. The inoculum was prepared by macerating CABMV-infected leaves in 10 mM sodium phosphate buffer, pH 7.0. The same procedure was used for mock inoculations of control plants with sodium phosphate buffer.

### 4.5. Greenhouse Experiments

In the UFRJ experiments (Rio de Janeiro, RJ), seedlings of ‘Redondo Amarelo’ and ‘H09-110/111’ were germinated in 9 L plastic pots, containing a commercial substrate (MECPLANT, Telêmaco Borba, Paraná, Brazil), and kept in a greenhouse under natural light conditions and a controlled temperature of 27 ± 2 °C. Three days after treatment, plants received mechanical CABMV inoculation. Leaf samples were collected for further analysis of defense-related and phytohormone-associated genes at 12 and 168 h after virus inoculation (hai), and a biomass evaluation was performed at 7 weeks after inoculation (wai). The experimental design was randomized biological replicates with 10 plants of ‘Redondo Amarelo’ and 4 plants of ‘H09-110/111’ for each treatment.

For greenhouse experiments conducted at EMBRAPA Agrobiologia (Seropédica, RJ), two genotypes of passion fruit seedlings, ‘FB300’ and ‘H09-110-111’, were grown in 15 L plastic pots containing 1/3 vermiculite substrate, 1/3 earthworm humus, and 1/3 washed sand, under natural temperature and light conditions of a tropical area. Three days after treatment, plants were mechanically inoculated with CABMV. Plants were distributed in a vertical trellis system with a wire 2.0 m above the plastic pot of each plant. Cover fertilizations were carried out every 10 days with worm-bed leachate (earthworm leachate solution) (1 L seedling^−1^) and castor bean cake (residue from the biodiesel production) (50 g seedling^−1^). Leaf samples were collected 24 and 72 h after treatment (hat) and morphological characters such as plant height and number of leaves were evaluated over time. The experimental design was randomized biological replicates, with 6 plants of ‘FB300’ and 9 plants of ‘H09-110/111’ for each treatment.

### 4.6. Field Experiments

To assess Hariman protection against CABMV in passion fruit plants under field conditions, an experiment was set up between March 2020 and July 2021 in the experimental fields of EMBRAPA Agrobiologia at Fazendinha Agroecológica (Seropédica, RJ). Passion fruit plants ‘FB300’ and ‘H09-110/111’ were grown in bags and kept in a greenhouse until the 3 to 4 true leaves stage, when they were foliar sprayed with our Hariman treatment. Three days after treatment, two leaves per plant were mechanically inoculated with CABMV. One week after CABMV inoculation (wai), plants were transplanted to the field, at a tree spacing of 2.5 × 1.0 m (4000 plants ha^−1^). In-field cultivation was conducted by espalier on number 12 wire threads fixed at 2.0 m above the ground. Pits (0.4 × 0.4 × 0.4 m) were fertilized with 150 g of thermophosphate (Yoorin^®^ Master 1, Yoorin, São Paulo, Brazil), 300 g of phonolytic rock powder (Yoorin^®^ eKoSil, Yoorin, São Paulo, Brazil), and 1 kg of corral manure. After planting, cover fertilizations were carried out three times every 45 days with corral manure (1 kg seedling^−1^) and castor bean cake (200 g seedling^−1^). A drip irrigation system was installed, using flow drippers (4 L hour^−1^ plant^−1^), activated for 1 h per day. The experimental design was in randomized blocks, with 4 treatments, 3 replicates, and 4 plants per plot (*n* = 12 plants per treatment). Treatments consisted of (a) Uninoculated; (b) Water-treatment + CABMV-inoculated; (c) Hariman-treatment + CABMV-inoculated (once sprayed); and (d) Hariman-treatment + CABMV (twice sprayed, with an interval of 12 weeks) plants.

The plants were monitored over time for the appearance of CABMV disease symptoms and/or the evaluation of plant morphological parameters.

Another experiment was carried out under field conditions in the municipality of Brumado (State of Bahia) between June and October 2023, with the same characteristics as the previous experiment, and with a plant spacing of 3 × 1.5 m (2222 plants ha^−1^). The genotype used was ‘BGP418’ from the EMBRAPA germplasm bank, and we evaluated the productivity effects of Hariman alone or in combination with other bioinducers such as plant growth promoting bacteria (PGPB). Hariman was foliar sprayed at a concentration of 100 μg.mL^−1^ and PGPB at a concentration of 106 CFU mL^−1^ in the nursery and field phases, with an interval of 60 days between applications. The PGPB used were *Bacillus* sp. BR 10433 and *Rhizobium* sp. BR 12157.

### 4.7. RT-qPCR Gene Expression

The expression of five defense-related genes *phenylalanine ammonia-lyase* (*PAL*), *peroxidase 12* (*POD12*), *lipoxygenase 2* (*LOX2*), *pathogenesis-related protein 3* (*PR-3*), and *superoxide dismutase* (*SOD*) [22, 46], and two genes involved in phytohormone signaling pathways such as *auxin-responsive protein SAUR20* (*SAUR20*) and *gibberellin 2-beta dioxygenase 2* (*GA2ox*) [30] were evaluated in all the Hariman-treated plants. Leaf samples were collected in biological triplicate (under greenhouse conditions) and quadruplicate (under field conditions) for gene expression analysis. Total RNA was extracted with TRIzol^®^ reagent (ThermoFisher Scientific, Waltham, MA, USA). SuperScriptTM VILOTM MasterMix (Invitrogen, ThermoFisher Scientific, Waltham, MA, USA) and PowerUpTM SYBRTM Green Master Mix (Thermo Fisher Scientific, Waltham, MA, USA) were used for cDNA synthesis and qPCR reactions, respectively. Three genes were used as internal controls for qPCR normalization: *NDID* (*NADP-dependent isocitrate dehydrogenase*), *ERS* (*ethylene response sensor*), and *EF1A1* (*translation elongation factor 1a-1*) [22,46]. qPCR of 3 independent biological pools were performed for each evaluated point in technical triplicates on Applied Biosystems 7500 Fast Real-Time PCR apparatus with the following cycling conditions: initial denaturation at 95 °C for 2 min, 40 cycles of 95 °C for 15 s, 52 –68 °C for 1 min, and elongation at 72 °C for 1 min. Ct values were evaluated using the 2^−ΔΔCt^ method and represented as fold change as proposed by [47].

### 4.8. Qualitative, Semi-Quantitative, and Quantitative Detection of CABMV

Young leaves from each treatment were collected at different times. Samples of ‘Redondo Amarelo’ plants were collected at 12 and 168 hai while samples of ‘FB300’ and ‘H09-110/111’ plants were collected at 4, 8, and 12 wai in the field.

For the qualitative RT-PCR diagnosis of CABMV, the total RNA from 12 of our passion fruit samples was extracted using TRIzol (Thermo Fisher Scientific, Waltham, MA, USA). RT-PCR was performed using a SuperScript III One-Step RT-PCR with the Platinum DNA Polymerase Kit (Invitrogen, Thermo Fisher Scientific, Waltham, MA, USA) and CABMV capsid-specific oligos, CABMV_M1 MX3726F 5′ GAGACACAAGCCAAAACACAAAATC 3′ and CABMV_M1 MX5029R 5′ CGTTGCTACAAATTCTGGTATCTCC 3′, which generate an expected amplicon of 1311 bp [9]. The amplified products were visualized under UV light in a 1% (*w*/*v*) agarose electrophoresis gel with ethidium bromide (0.5 µg.mL^−1^) (Promega, Madison, WI, USA).

PathoScreen^®^ (Agdia, Elkhart, IN, USA) ELISA kit for specific detection of viruses of the potyvirus group including CABMV was used to identify the presence and relative accumulation (Absorbance at OD415) of CABMV in passion fruit plants following the Agdia protocol. A total of 12 samples per treatment (*n* = 12) were analyzed.

The RT-qPCR reactions to quantify CABMV were performed using the primers qCABMV06_For 5′ ATAGAATACAAGCCAGCACAAATCG 3′ and qCABMV06_Rev 5′ CCGTCCATCATAGTCCACACC 3′, designed to amplify a 200 pb fragment of CABMV coat protein gene [8]. Normalization of Ct values was performed using the constitutive primers described in the section ‘RT-qPCR gene expression’. Three biological samples were evaluated per treatment and time after CABMV infection. All reactions were conducted in technical triplicates as follows: incubation at 50 °C for 2 min, activation of Taq DNA polymerase at 95 °C for 10 min, followed by 40 denaturation cycles at 95 °C for 15 s, annealing of the primers 55 °C, and extension at 60 °C for 1 min. Ct values were evaluated using the 2^−ΔΔCt^ method as proposed by [48].

### 4.9. Evaluation of the Effect of Hariman in Development and Productivity

The fresh and dry biomass of the aerial parts and root system of greenhouse-grown passion fruits ‘H09-110/111’ were evaluated for each treatment at 7 wai. An electronic scale (Bioprecisa—JA3003N, Bioprecisa, Curitiba, PR, Brazil) was used to measure biomass weight. The samples were weighed after washing and wiped dry for the determination of fresh weight before the samples were oven-dried at 65 °C for 72 h for the dry weight. For passion fruits ‘FB300’ and ‘H09-110/111’, developmental characters such as height (estimated with a measuring tape) and the number of leaves over time were evaluated.

For plants growing under field conditions, the development and productivity characteristics evaluated were height, number of leaves, number of flower buds, and number of fruits. The last fruit harvest was at 43 wai. In total, 30 fruits per each treatment were used to evaluate the fresh weight (WF, in g); pulp weight (WP, in g); yield of fruit pulp (YP, calculated through the difference between WP and WF); soluble solids (SS, in °Brix); and total yield (TY, in tons ha^−1^). WF and WP were measured using a digital scale (Bioprecisa—JA3003N, Bioprecisa, Curitiba, PR, Brazil) and SS was measured using a portable sucrose refractometer with 0–53 °Brix scale (PAL-1, ATAGO^®^, Tokyo, Japan). In addition, the number of fruits produced per plant was checked at three different times prior to harvesting and counting only those fruits present at about 4 weeks after blooming. Fruit production per plant (in kg) was calculated by multiplying the average number of fruits per plant (TNF) by the mean WF for the respective treatment. Finally, individual plant production was extrapolated on a per-hectare basis as a function of the number of plants per hectare for estimating yield in t ha^−1^.

### 4.10. CABMV Incidence and Severity

CABMV incidence and severity were estimated, taking into account typical leaf symptoms of the virus. Data were collected between 2 and 9 weeks after CABMV inoculation (wai) and disease incidence (DI%) was determined based on the symptoms on diseased plants (with at least one leaf showing mild to severe mosaic and/or deformation). The proportion of diseased plants was estimated by DI = (*n*/N) × 100 (DI = incidence; *n* = number of diseased plants; N = total number of plants assessed) [49]. Disease severity (DS%) was calculated using a symptom severity rating scale of 1, 2, 3, and 4 where 1 represented the absence of symptoms, 2 represented the presence of mild mosaic symptoms without leaf deformation, 3 represented the presence of severe mosaic without leaf deformation, and 4 represented the presence of severe mosaic, blisters, and leaf deformation, as proposed by [50]. DS index was then determined for each treatment using the formula according to Mckinney [47] as shown below: DS = ∑(DS × L)/(TNP × HGS) × 100, where DS = degree of the 1–4 scale determined for each plant; L = number of leaves showing each degree of infection (score); TNP = total number of leaves evaluated; and HGS = highest grade of the scale (maximum infection score). Six to nine plants per treatment were evaluated in the greenhouse, and twelve plants per treatment were evaluated in the field.

### 4.11. Data Analysis

Data obtained from gene expression, disease incidence (DI), development, productivity, and yield parameters were analyzed using parametric tests. For comparisons involving multiple groups, a two-way ANOVA was applied to assess the effects of Hariman treatment and CABMV inoculation, while a One-way ANOVA was used for single-factor comparisons. Both tests were followed by the Bonferroni post hoc test (*p* < 0.05) to correct for multiple comparisons and control the family-wise error rate, ensuring robust statistical conclusions. The Bonferroni correction was particularly important to minimize the risk of Type I errors (false positives) given the number of comparisons performed across treatments and time points. For CABMV accumulation data obtained from RT-qPCR, a Student’s *t*-test was used to compare means between two groups, with a significance level of *p* < 0.05. For disease severity (DS) and ELISA data, which did not meet parametric assumptions, a non-parametric Kruskal–Wallis test was applied, followed by Dunn’s post hoc test (*p* < 0.05) to control for multiple comparisons. All statistical analyses were performed using GraphPad Prism software version 5.00, and results are presented as means ± standard deviation (SD). Differences were considered statistically significant at *p* < 0.05.

## 5. Conclusions

The results of this study highlight the significant potential of the organic biostimulant Hariman in enhancing passion fruit cultivation by effectively mitigating CABMV virus infections. Our findings demonstrate that Hariman not only reduces the incidence and severity of viral infections but also promotes overall plant health and productivity. The treatment enhances defense mechanisms, as evidenced by the sustained expression of defense-related genes, reduced viral accumulation, and improvement in morphological traits and productivity parameters, indicating its critical role in modulating phytohormonal signaling pathways. The observed increases in fruit yield and quality, combined or not with PGPB, underscore the effectiveness of these treatments in achieving superior agricultural outcomes. Our study provides compelling evidence for the application of Hariman as a sustainable strategy to enhance resilience against viral pathogens in passion fruit and potentially other crops facing similar challenges. Further research is encouraged to explore the broader implications of biostimulant applications in diverse agricultural systems.

## 6. Patents

Granted patent BR 102017002045-2, INPI, Brazil and Patent BR 102020026358-7.

## Figures and Tables

**Figure 1 ijms-26-02177-f001:**
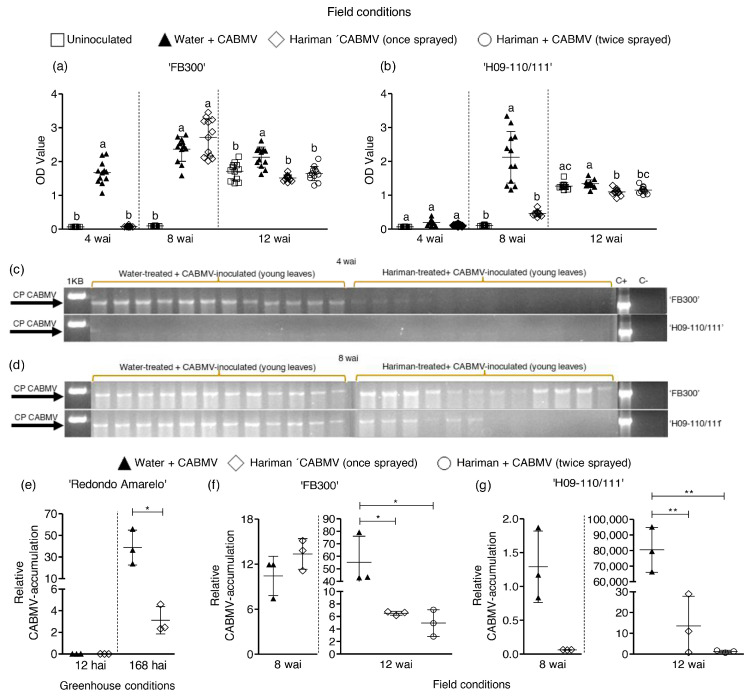
Effect of Hariman on relative CABMV accumulation in passion fruit plants under greenhouse and field conditions. Cowpea aphid-borne mosaic virus (CABMV) was detected in tissue samples of CABMV-inoculated *P. edulis* ’FB300’ (**a**) and ‘H09-110/111’ (**b**) plants treated with water and Hariman by ELISA using standardized optical density (OD) values. The ELISA tests were conducted 4, 8, and 12 weeks after inoculation (wai). CABMV presence was also assayed by RT-PCR in passion fruit plants under field conditions. Young leaf samples from ‘FB300’ and ‘H09-110/111’ plants treated with Hariman + CABMV and water + CABMV were collected at 4 wai (**c**) and at 8 wai (**d**). Arrows indicate the amplification of the 1311 bp fragment, confirming the presence of the CABMV coat protein (CP). The amplified products were separated by 1% (*w*/*v*) agarose gel electrophoresis, stained with ethidium bromide, and visualized under ultraviolet (UV) light. The 1 kb DNA Ladder Plus (LabAid™, Invitrogen, Thermo Fisher Scientific, Waltham, Massachusetts, USA) was used as a size marker. C+ indicates the CABMV positive control, and C− indicates the negative control. Additionally, RT-qPCR results indicate relative CABMV accumulation in ‘Redondo Amarelo’ at 12 and 168 h after inoculation (hai) under greenhouse conditions (**e**), and in ‘FB300’ (**f**) and ‘H09-110/111’ (**g**) at 8 and 12 wai under field conditions. Ct values were normalized to reference genes *ERS*, *NDID*, and *EF1a1*. Statistically significant differences are indicated by bars with different letters or asterisks. * *p* < 0.05 and ** *p* < 0.01, respectively. Individual data points are represented by dots, with the horizontal line within the box denoting the average value. Whiskers extend to the maximum and minimum values within the interquartile range.

**Figure 2 ijms-26-02177-f002:**
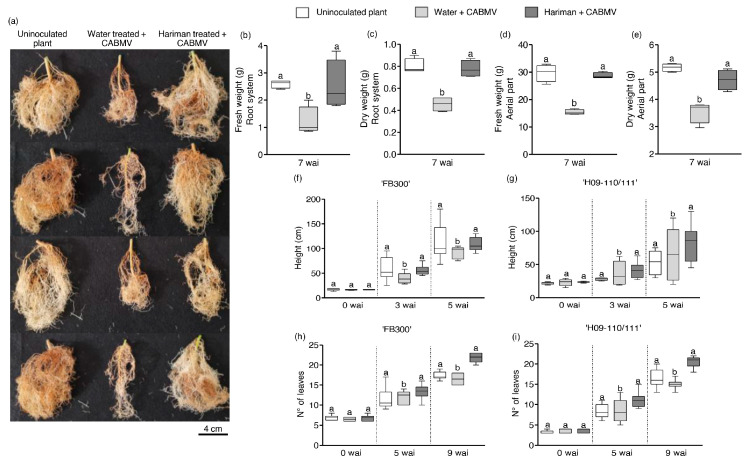
Effect of Hariman on developmental parameters of passion fruit plants under greenhouse conditions. Panel (**a**) shows the root system morphology of ‘H09-110/111’ plants at 7 weeks after CABMV inoculation. Panels (**b**–**e**) present biomass analysis of the root system (fresh and dry weight (**b**,**c**)) and aerial parts (**d**,**e**) for ‘H09-110/111’. Panels (**f**–**i**) display development parameters, including plant height at 0–5 weeks after inoculation (wai) for ‘FB300’ (**f**) and ‘H09-110/111’ (**g**), and number of leaves at 0–9 wai for ‘FB300’ (**h**) and ’H09-110/111’ (**i**). Data are represented as the mean ± standard error of the mean (SEM) from independent experiments with 4 plants per treatment (**b**), 9 plants per treatment (**f**,**h**), and 6 plants per treatment (**g**,**i**). Different letters indicate significant differences (*p* < 0.05). Box plots display the first and third quartiles with the average represented by the horizontal line within the box. Whiskers extend to the maximum and minimum values within the interquartile range.

**Figure 3 ijms-26-02177-f003:**
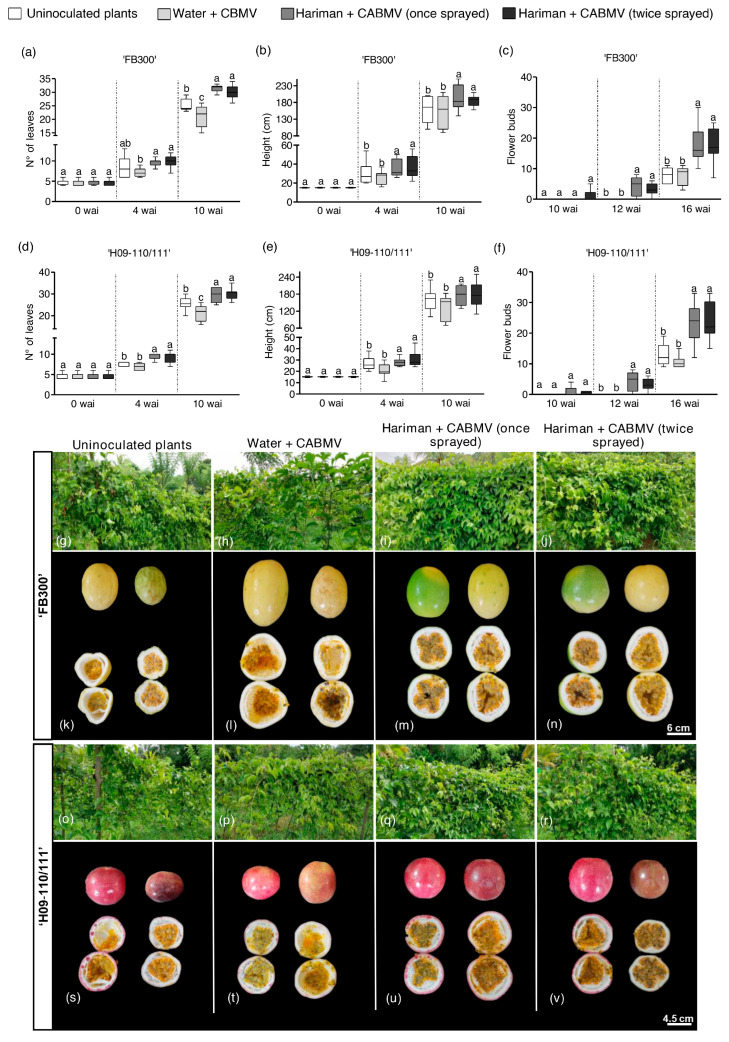
Effect of Hariman on developmental parameters in passion fruit plants under field conditions. Panels (**a**–**c**) show the effect of Hariman on ‘FB300’ passion fruit plants at 0–10 weeks after inoculation (wai), including the number of leaves (**a**), plant height (**b**), and flower buds (**c**) at 10–16 wai. Panels (**d**–**f**) depict the same parameters for ‘H09-110/111’ plants. Panels (**g**–**v**) show the development of whole passion fruit plants for both cultivars at 25 wai. Panels (**k**–**n**) and (**s**–**v**) show two representative samples of the overall appearance of the fruits obtained under all the experiments from ‘FB300’ and H09-11-/111’ plants, respectively. Bars represent the average of independent replicates (*n* = 12 plants per treatment), and different letters indicate significant differences (*p* < 0.05). Box plots show the first and third quartiles with the average represented by the horizontal line inside the box. Whiskers extend to the maximum and minimum values within the interquartile range.

**Figure 4 ijms-26-02177-f004:**
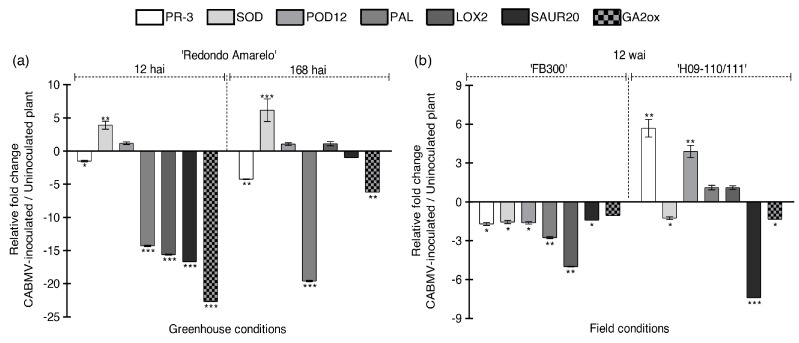
CABMV infection drastically affect expression of defense-related and phytohormone signaling pathway genes of passion fruit plants under greenhouse and field conditions. RT-qPCR results showing relative fold change in gene expression of defense-related genes (*PR-3*, *SOD*, *POD12*, *PAL*, and *LOX2*) and phytohormone signaling pathway genes (*SAUR20* and *GA2ox*) in CABMV-inoculated plants versus uninoculated controls. Panel (**a**) shows data for ‘Redondo Amarelo’ plants at 12 and 168 h after inoculation (hai) under greenhouse conditions, while panel (**b**) shows data for ‘H09-110/111’ and ‘FB300’ plants at 12 weeks after inoculation (wai) under field conditions. Ct values were normalized to *ERS*, *NDID*, and *EF1a1*. Data are presented as the average relative fold change in mRNA transcripts from three technical replicates of three biologically independent replicates for each treatment and time point. Error bars represent standard deviation. Statistical significance is indicated by * (*p* < 0.05), ** (*p* < 0.01), and *** (*p* < 0.001).

**Figure 5 ijms-26-02177-f005:**
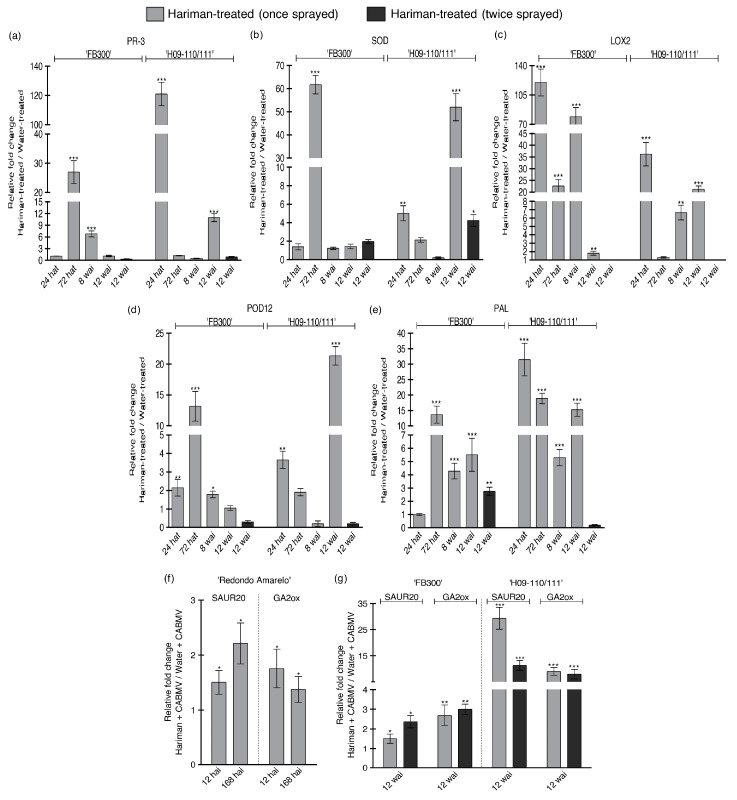
Hariman-induced expression of defense-related and phytohormone signaling pathway genes in CABMV-inoculated passion fruit plants under field and greenhouse conditions. Panels (**a**–**e**) show RT-qPCR relative fold change in gene expression of defense-related genes PR-3 (**a**), SOD (**b**), LOX2 (**c**), POD12 (**d**), and PAL (**e**) in ‘FB300’ and ‘H09-110/111’ passion fruit plants at 24 and 72 h after treatment (hat) under greenhouse conditions 8 and 12 weeks after inoculation (wai) under field conditions compared with Hariman + CABMV (once and/or twice sprayed) to water + CABMV treatment. Panel (**f**) presents gene expression of phytohormone signaling pathway genes *SAUR20* and *GA2ox* in ‘Redondo Amarelo’ plants at 12 and 168 h after inoculation (hai) under greenhouse conditions. Panel (**g**) shows expression of the same phytohormone signaling pathway genes (*SAUR20*, *GA2ox*) in ‘FB300’ and ‘H09-110/111’ at 12 wai under field conditions. Gene expression is expressed as the relative fold change in Hariman + CABMV (once and/or twice sprayed) compared with water + CABMV. Ct values were normalized to *ERS*, *NDID*, and *EF1a1*. Data are presented as the average relative fold change in mRNA transcripts from three technical replicates of three biologically independent replicates for each treatment and time point. Error bars represent standard deviation. Statistical significance is indicated by * (*p* < 0.05), ** (*p* < 0.01), and *** (*p* < 0.001).

**Table 1 ijms-26-02177-t001:** Hariman treatment effect on the CABMV disease incidence (DI) and disease severity (DS) in passion fruit plants of ‘FB300’ (a) and ‘H09-110/111’ (b) in field conditions through the time.

**(a)** **‘FB300’**	**DI**				**DS**			
**Treatments**	**4 Wai**	**5 Wai**	**7 Wai**	**9 Wai**	**4 Wai**	**8 Wai**	**12 Wai**	**16 Wai**
Uninoculated	50.0 ± 52.2 ab	58.3 ± 51.4 a	83.3 ± 38.9 a	100 ± 0.0 a	31.2 ± 33.9 ab	78.1 ± 16.1 a	81.0 ± 19.1 a	73.5 ± 15.6 ab
Water + CABMV	91.6 ± 28.8 a	100 ± 0.0 a	100 ± 0.0 a	100 ± 0.0 a	66.6 ± 28.8 a	83.3 ± 14.2 a	87.7 ± 18.1 a	79.1 ± 13.7 a
Hariman + CABMV	25.0 ± 45.2 b	58.3 ± 51.4 a	83.3 ± 38.9 a	100 ± 0.0 a	17.7 ± 33.0 b	56.2 ± 30.3 b	71.2 ± 20.3 a	60.0 ± 16.1 b
Hariman 2× + CABMV	33.3 ± 49.2 b	58.3 ± 51.4 a	83.3 ± 38.9 a	100 ± 0.0 a	22.9 ± 34.4 b	51.0 ± 26.9 b	78.7 ± 19.2 a	60.6 ± 14.5 b
*p* value (*n* = 12)	* 0.0035	* 0.0627	* 0.5377	* 0.5770	** 0.0068	** 0.0015	** 0.1160	** 0.0055
**(b)** **‘H09-110/111’**	**DI**				**DS**			
Uninoculated	33.3 ± 49.2 b	50.0 ± 52.2 ab	83.3 ± 38.4 a	100 ± 0.0 a	18.5 ± 29.7 b	51.9 ± 26.4 b	70.0 ± 17.7 a	60.7 ± 15.4 a
Water + CABMV	91.6 ± 28.8 a	100 ± 0.0 a	100 ± 0.0 a	100 ± 0.0 a	68.7 ± 34.3 a	86.4 ± 11.2 a	80.4 ± 18.1 a	67.0 ± 22.9 a
Hariman + CABMV	33.3 ± 49.2 b	41.6 ± 51.4 b	75.5 ± 45.2 a	100 ± 0.0 a	21.8 ± 33.7 b	63.5 ± 25.2 b	61.6 ± 20.1 a	55.2 ± 14.4 a
Hariman 2× + CABMV	41.6 ± 51.4 b	41.6 ± 51.4 b	76.5 ± 43.2 a	100 ± 0.0 a	29.1 ± 36.6 b	57.2 ± 33.0 b	63.9 ± 18.2 a	54.5 ± 10.9 a
*p* value (*n* = 12)	* 0.0074	* 0.0062	* 0.3248	* 0.5770	** 0.0032	** 0.0021	** 0.0811	** 0.5676

Each value is the means (±SD) of an independent experiment. Different letters in columns indicate significant differences between treatments. * Corresponding results of One-Way ANOVA. Significant differences according to using the Bonferroni post hoc test (*p* < 0.05). ** The *p* value is based on a non-parametric Kruskal–Wallis test. Significant Dunn’s multiple comparison test (*p* < 0.05).

**Table 2 ijms-26-02177-t002:** The effect of Hariman on the estimated productivity of passion fruit ‘FB300’ (a) and ‘H09-110/111’ (b) in field conditions. Total number of fruits (TNF), total yields (TY), weight fruit (WF), weight pulp (WP), pulp yield (PY), soluble solids content—°Brix (SS), length (LF), diameter (DF).

**‘FB300’**						
**Treatments**	**TNF**	**TY (Tons ha^−1^)**	**WF (g)**	**WP (g)**	**PY (%)**	**SS (°Brix)**
Uninoculated	10.83 ± 1.46 b	7.54 ± 1.47 c	161.8 ± 53.92 b	70.85 ± 27.89 b	43.34 ± 8.64 a	13.73 ± 2.53 a
Water-treated + CABMV	10.50 ± 5.46 b	7.27 ± 2.21 c	168.0 ± 62.33 b	58.87 ± 27.92 b	34.40 ± 12.54 b	14.88 ± 1.71 a
Hariman-treated + CABMV	16.42 ± 4.68 a	12.05 ± 2.32 b	215.6 ± 81.57 a	99.96 ± 36.43 a	46.66 ± 9.62 a	13.59 ± 1.74 a
Hariman-treated 2× + CABMV	20.67 ± 3.79 a	15.58 ± 2.90 a	217.7 ± 70.58 a	101.5 ± 37.71 a	46.49 ± 3.43 a	14.35 ± 1.18 a
*p* value *	0.0001	0.0001	0.0001	0.0001	0.0001	0.0311
**‘H09-110/111’**						
**Treatments**	**TNF**	**TY (Tons ha^−1^)**	**WF (g)**	**WP (g)**	**PY (%)**	**SS (°Brix)**
Uninoculated	11.83 ± 3.38 a	7.43 ± 1.01 c	158.9 ± 24.52 a	60.77 ± 13.17 b	38.05 ± 4.74 c	14.15 ± 1.76 a
Water-treated + CABMV	13.67 ± 4.25 ab	8.61 ± 1.70 c	145.5 ± 31.20 ac	55.94 ± 15.57 b	41.41 ± 9.61 bc	15.12 ± 1.87 a
Hariman-treated + CABMV	17.50 ± 6.44 bc	12.44 ± 1.51 b	188.7 ± 54.08 ab	78.07 ± 26.16 a	47.22 ± 4.30 a	14.45 ± 1.46 a
Hariman-treated 2× + CABMV	19.42 ± 3.28 c	15.50 ± 2.36 a	174.2 ± 45.60 ab	66.74 ± 24.15 ab	43.93 ± 6.09 ab	15.19 ± 1.13 a
*p* value *	0.0006	0.0001	0.0005	0.0004	0.0001	0.0293

Each value is the means (±standard deviation) of one experiment using 12 plants per treatment. The averages of WF, WP, PY, SS, LF, and DF use 30 fruits per treatment. Different letters in columns indicate significant differences between treatments according to using the Bonferroni post hoc test (*p* < 0.05). * Corresponding results of One-Way ANOVA.

## Data Availability

All data are present in this document and/or in the Appendix A.

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
