# Peer review of "Enhancing Passion Fruit Resilience: The Role of Hariman in Mitigating Viral Damage and Boosting Productivity in Organic Farming Systems"

_ijms, 2025, doi:10.3390/ijms26052177_

Round 1
Reviewer 1 Report
Comments and Suggestions for Authors
This article reports the antiviral effect of Hariman @ in passion fruit and explores its molecular mechanism. I am surprised that its antiviral effect is very strong, and after using it twice, it can increase production. I suggest publishing this interesting paper in this journal. Here are some of my suggestions for authors to refer to:
1. Why did this research choose Hariman@ as an antiviral method?
2. Why only choose PR-3 (a), SOD (b), LOX2 (c), POD12 (d), and PAL (e) as the five defense mechanisms for the article to explore its antiviral mechanism?
3. These five defense mechanisms have some emphasis on their roles, such as PR-3 expression being directly related to antiviral effects, PAL being closely related to secondary metabolites that play a defensive role in plants, and the other three being closely related to antioxidant effects. What are the connections between these three mechanisms, and how do they work together to exert their ultimate antiviral and even yield increasing effects?
4. In Figure 1, e - g, n=3, not 12, right?
5. If you are confident in the conclusion, I suggest indicate the key "defense genes and phytohormone" in the abstract section?
Author Response
- Reviewer 1
This article reports the antiviral effect of Hariman in passion fruit and explores its molecular mechanism. I am surprised that its antiviral effect is very strong, and after using it twice, it can increase production. I suggest publishing this interesting paper in this journal. Here are some of my suggestions for authors to refer to:
Authors: Thank you for all comments and suggestions. They will improved paper quality.
- Comment 1. Why did this research choose Hariman as an antiviral method?
Response 1: Previous research by our group showed that the pGM, the active principle of Hariman, induced around 50% of reduction on the number of necrotic lesions caused by tobacco mosaic virus (TMV) on tobacco infected plants (Montebianco, CB, PhD Thesis 2021 UFRJ). Besides that, another previous work has shown that the contact for some minutes with pGM induces the expression of defense and SAR related genes strongly in BY2 tobacco cells (Mattos et al., 2018). We also showed that pGM is able to protect passion fruit against CABMV in greenhouse experiments (Santos-Jiménez et al., 2022a). So, our previous work prompted us to go further and test the antiviral effect of Hariman in field conditions.
- Comment 2. Why only choose PR-3 (a), SOD (b), LOX2 (c), POD12 (d), and PAL (e) as the five defense mechanisms for the article to explore its antiviral mechanism?
Response 2: In a previous paper of the group we showed that pGM/Hariman contact was able to induce PR-1, PR2, PR-3, peroxidase, lipoxygenase and PAL in tobacco cells (Mattos et al., 2018). The same was observed when tobacco plants were sprayed with pGM (Montebianco, CB, PhD Thesis 2021 UFRJ and Montebianco et al, in preparation). So, we tried to look for these genes and other defense related genes in passion fruit…
The choice of these genes was thought to check if SAR was induced after pGM foliar spray in passion fruit (POD12, PR-3, PAL) and/or to check antiviral activity induction that is generally associated with LOX, POD12 and SOD. For sure several other genes could be included in the expression profile analysis, however these genes are classically markers of biotic defense induction.
During stress mediated by pathogen attack, ROS is activated in plant tissue, which damages internal cellular structures and induces ROS control activation. We showed previously that pGM (that is a fungal cell wall component and one of the first molecules to contact plant cell during fungus interaction) induced SAR and local HR in tobacco leaves (Mattos et al., 2018). So we checked if elicitation with Hariman will also induce HR in passion fruit and a counter response against ROS induction. Another key mechanism of plant response against pathogen attack is the production of secondary metabolites, antioxidants and phenols, so we looked for the key enzyme of phenol metabolism, PAL, in the Hariman treated plants. As well on the CABMV infection in passion fruit plants.
- Comment 3. These five defense mechanisms have some emphasis on their roles, such as PR-3 expression being directly related to antiviral effects, PAL being closely related to secondary metabolites that play a defensive role in plants, and the other three being closely related to antioxidant effects. What are the connections between these three mechanisms, and how do they work together to exert their ultimate antiviral and even yield increasing effects?
Response 3: We sincerely appreciate the reviewer’s insightful comment. The defense mechanisms induced by Hariman are probably much bigger than we sought in this work. However, in the picture we have, it is clear that the defense mechanisms mediated by PR-3, PAL, SOD, POD12, and LOX2 are activated by the elicitor Hariman. These genes seem to function in a coordinated manner to enhance antiviral resistance. PR-3 encodes a chitinase that directly disrupts viral structures, limiting replication and movement within the plant, ultimately reducing viral load and symptom severity (Dos Santos & Franco, 2023). PAL, a key enzyme in the phenylpropanoid pathway, facilitates the biosynthesis of secondary metabolites such as flavonoids and lignin, which strengthen physical and chemical barriers against pathogens and contribute to overall plant resilience (Kaur et al., 2022). Meanwhile, the antioxidant defense system plays a crucial role in mitigating oxidative stress caused by viral infection. SOD catalyzes the conversion of superoxide radicals into hydrogen peroxide (H₂O₂), which is subsequently decomposed by POD12, preventing cellular damage and maintaining redox homeostasis (Munhoz et al., 2015). LOX2, on the other hand, is involved in the biosynthesis of jasmonic acid, a central regulator of defense responses against biotic stressors, including viral infections (Roychowdhury et al., 2024). We thought that together these mechanisms establish a robust, multi-layered defense strategy, where PR-3 directly inhibits viral progression, PAL reinforces structural integrity and induces the accumulation of all the phenol pathway derived products, and SOD, POD12, and LOX2 regulate reactive oxygen species (ROS) levels and stress-related signaling pathways.
In addition to these defense responses, plant-virus interactions often disrupt growth by rapidly altering phytohormone levels and their signaling pathways. Viral infections have been shown to interfere with plant defense by promoting conditions that favor viral replication and systemic spread. Our results revealed that untreated plants exhibited suppression of SAUR20 and GA2ox, genes involved in auxin and gibberellin metabolism, respectively, which is consistent with previous findings on virus-induced hormonal disruption (Chen et al., 2021). However, Hariman-treated plants maintained normal expression levels of these genes even after CABMV inoculation, suggesting that Hariman primes the plant’s defense in a way that prevents viral interference with phytohormonal regulation. As a result, treated plants displayed normal vegetative growth, preserved root system integrity, and sustained aerial development, which contributed to increased flowering and fruiting rates, ultimately leading to higher crop yields. Our results demonstrated significant productivity gains in treated plants (65.7% and 114% for 'FB300' and 44% and 80% for 'H09-110/111' after one and two applications, respectively), further reinforcing the idea that Hariman not only mitigates viral stress but also enhances physiological processes critical for plant performance.
In summary, the interplay between direct antiviral action (PR-3), structural reinforcement and secondary metabolites production (PAL), oxidative stress regulation (SOD, POD12, LOX2), and the preservation of phytohormonal homeostasis allows Hariman-treated plants to mount a strong defense against CABMV while maintaining optimal growth and yield. This comprehensive protection enables plants to withstand viral infection without suffering developmental trade-offs, ensuring both resilience and enhanced productivity. We appreciate the opportunity to clarify these points and hope this explanation fully addresses the reviewer’s concerns.
- Comment 4. In Figure 1, e - g, n=3, not 12, right?
Response 4: We thank the reviewer for their observation regarding the sample size in Figure 1e-g. We would like to clarify that for the RT-qPCR experiments, we used pools of 3 biological independent samples with technical triplicates, rather than 12 individual samples as in the ELISA (Figure 1a-b) and RT-PCR (Figure 1c-d) assays. This approach was chosen to ensure sufficient RNA quantity and quality for gene expression analysis while maintaining statistical robustness. We apologize for any confusion and have revised the manuscript to clearly specify the sample sizes and experimental design for each assay. Thank you for bringing this to our attention.
- Comment 5. If you are confident in the conclusion, I suggest indicate the key "defense genes and phytohormone" in the abstract section?
Response 5: Thank you for your comment. The induction of the analyzed defense gene and phytohormones is very robust and persists during distinct time points in both passion fruit cultivars. Besides that, the levels of induction were very high in most of the time points analysed. We have indicated the genes in the abstract.
References:
- Chen, L., Sun, D., Zhang, X., Shao, D., Lu, Y., & An, Y. (2021). Transcriptome analysis of yellow passion fruit in response to cucumber mosaic virus infection. PLoS One, 16(2), e0247127.
- Dos Santos, C., & Franco, O. L. (2023). Pathogenesis-related proteins (PRs) with enzyme activity activating plant defense responses. Plants, 12(11), 2226.
- Kaur, S., Samota, M. K., Choudhary, M., Choudhary, M., Pandey, A. K., Sharma, A., & Thakur, J. (2022). How do plants defend themselves against pathogens-Biochemical mechanisms and genetic interventions. Physiology and Molecular Biology of Plants, 28(2), 485-504.
- Mattos, BB; Montebianco, CB; Romanel, E; Silva, T da F; Bernabé, RB; Simas-Tosin, F; Souza, LM; Sassaki, GL; Vaslin, MFS; Barreto-Bergter, E. A peptidogalactomannan isolated from Cladosporium herbarum induces defense-related genes in BY-2 tobacco cells. Plant Phys. Biochem, v.126. p. 206-216, 2018.
- Munhoz, C. F., Santos, A. A., Arenhart, R. A., Santini, L., Monteiro‐Vitorello, C. B., & Vieira, M. L. C. (2015). Analysis of plant gene expression during passion fruit–Xanthomonas axonopodis interaction implicates lipoxygenase 2 in host defence. Annals of Applied Biology, 167(1), 135-155.
- Roychowdhury, R., Hada, A., Biswas, S., Mishra, S., Prusty, M. R., Das, S. P., ... & Sarker, U. (2024). Jasmonic Acid (JA) in Plant Immune Response: Unravelling Complex Molecular Mechanisms and Networking of Defence Signalling Against Pathogens. Journal of Plant Growth Regulation, 1-26.
- Santos-Jiménez, J. L., de Barros Montebianco, C., Vidal, A. H., G Ribeiro, S., Barreto-Bergter, E., & Vaslin, M. F. S. (2022a). A fungal glycoprotein mitigates passion fruit woodiness disease caused by C owpea aphid-borne mosaic virus (CABMV) in Passiflora edulis. BioControl, 1-13.

Reviewer 2 Report
Comments and Suggestions for Authors
I acknowledge the overall quality and research content of this article, but I have some suggestions for improvements to further enhance its clarity, scientific rigor, and readability. Here are my specific revision suggestions:
- Some of the cited literature is rather outdated (around 2010, line 51), it is suggested to supplement with the latest research results from the past five years, especially regarding the application of CABMV and biostimulants in plant antiviral defense.
- In Materials and Methods section, it is recommended to add detailed explanations of the specific timing and application frequency of Hariman treatment, especially regarding the application intervals and dosage adjustments in field experiments.
- In Materials and Methods section, although statistical methods are mentioned, it can be further elucidated how data variability (such as high SEM values) is handled and whether multiple comparison corrections (such as Bonferroni correction) have been performed.
- Specific OD value ranges can be added to the ELISA results in Figure 1.
- In the high SEM value for chlorophyll content in Figure 3, the authors should discuss possible sources of experimental errors (such as individual plant differences, measurement methods, etc.) and explaining how to reduce this variability in the discussion.
- It is recommended to add a mechanism discussion of the effect of Hariman treatment , especially the correlation in gene expression and virus accumulation, to enhance the logicality of the results. While the impact of Hariman on defense genes and plant hormones has been discussed, further exploration can be done on how these genes and hormones synergistically enhance the antiviral ability of plants. For example, the specific mechanisms of action of the SA and JA signaling pathways under Hariman treatment can be discussed. It is recommended to increase the discussion on the limitations of the research, such as the long-term stability of the handling effects by Hariman, the applicability under different environmental conditions, etc.
- Some terms are inconsistently used, such as "Hariman" sometimes written as "Hariman®". It is recommended to unify it as "Hariman".
- Although the language of the article is generally fluent, there are still some minor errors, such as "phytohormones elicitation" should be changed to "phytohormone elicitation". It is recommended to conduct a comprehensive grammar and spelling check.
Author Response
Thank you for your suggestions and comments. They will improve MS quality.
Comment 1: Some of the cited literature is rather outdated (around 2010, line 51), it is suggested to supplement with the latest research results from the past five years, especially regarding the application of CABMV and biostimulants in plant antiviral defense.
Response1: We agree with this comment. Literature cited was updated with the last research results. New introduced references are highlighted in red.
Comment 2: In Materials and Methods section, it is recommended to add detailed explanations of the specific timing and application frequency of Hariman treatment, especially regarding the application intervals and dosage adjustments in field experiments.
Response 2: We appreciate the reviewer’s suggestion to provide more detailed information on the specific timing and application frequency of Hariman treatment, especially in field experiments. In response to this, we have now included further clarification on the application intervals and dosage adjustments in the Materials and Methods section. The Hariman treatment was applied at a concentration of 100 μg.mL-1 to passion fruit plants when they reached the 3 to 4 true leaves stage. For field experiments, a second application was performed 60 days after the first treatment for experiments conducted in Seropedica, RJ and with an interval of 12 weeks between applications for experiments conducted in Bahia. The plants were sprayed using a costal (back) manual sprayer (Jacto - XP) to ensure uniform coverage. We hope this additional information will clarify the treatment protocols for the field experiments.
The updated details on Hariman application timing and frequency have been included in the Materials and Methods section (see section 4.3, line 620 to 626, in red).
Comment 3: In Materials and Methods section, although statistical methods are mentioned, it can be further elucidated how data variability (such as high SEM values) is handled and whether multiple comparison corrections (such as Bonferroni correction) have been performed.
Response 3: We thank the reviewer for their observation regarding the need to clarify how data variability was handled and whether multiple comparison corrections were applied. In response, we have revised the Data analysis (seccion 4.11, line 774 to 789, in red) section to provide a more detailed and justified explanation of the statistical methods used. Specifically, we now emphasize that results are presented as means ± standard deviation (SD), which clearly reflects data variability within each treatment group. To address multiple comparisons, we applied the Bonferroni correction following Two-way and One-way ANOVA tests, ensuring control of the family-wise error rate and minimizing the risk of Type I errors (false positives). For non-parametric data (disease severity and ELISA), we used the Kruskal-Wallis test followed by Dunn’s post-hoc test, which is appropriate for such data while maintaining statistical rigor. All analyses were performed using GraphPad Prism version 5.00, a widely recognized software for statistical analysis. These updates demonstrate our rigorous handling of data variability and appropriate application of multiple comparison corrections, ensuring the robustness and validity of our conclusions. We hope these clarifications address the reviewer’s concerns.
Comment 4: Specific OD value ranges can be added to the ELISA results in Figure 1.
Response 4: We appreciate the reviewer’s suggestion regarding the inclusion of specific optical density (OD) value ranges for the ELISA results presented in Figure 1. In response to this comment, we have added a new supplementary table (Table S3) to the manuscript, which provides the detailed OD values for CABMV detection in passion fruit plants under greenhouse and field conditions. This table includes the individual OD values for each treatment group (uninoculated, water-treated + CABMV, Hariman-treated + CABMV, and Hariman-treated x2 + CABMV) at 4, 8, and 12 weeks after inoculation (wai) for both 'FB300' and 'H09-110/111' genotypes. The table also includes the average OD values and standard deviations (SD) for each treatment, along with the results of the statistical analysis (non-parametric Kruskal-Wallis test followed by Dunn’s multiple comparison test, p < 0.05). These data complement the information presented in Figure 1 and provide a more comprehensive view of the relative CABMV accumulation in response to Hariman treatment.
We hope that the addition of Table S3 addresses the reviewer’s concern and enhances the transparency and reproducibility of our results. Thank you for this valuable suggestion, which has improved the clarity of our manuscript.
Comment 5: In the high SEM value for chlorophyll content in Figure 3, the authors should discuss possible sources of experimental errors (such as individual plant differences, measurement methods, etc.) and explaining how to reduce this variability in the discussion.
Response 5: We appreciate the reviewer’s observation regarding the variability in the data presented in Figure 3. However, we would like to clarify that the study did not include chlorophyll content measurements; instead, Figure 3 focuses on developmental parameters such as number of leaves, plant height, and flower buds in passion fruit plants under field conditions. Regarding the variability observed in these parameters, we acknowledge that factors such as individual plant differences and small differences in environmental conditions can contribute to data dispersion. To minimize this variability, we implemented several measures:
- Biological Replicates: We used a sufficient number of independent replicates (n = 12 plants per treatment) to ensure statistical robustness and account for natural variability among plants.
- Standardized Measurements: All developmental parameters were measured using consistent methods (e.g., measuring tape for plant height, manual counting for leaves and flower buds) to reduce experimental error.
- Experimental Design: The study was conducted in a randomized block design to control for environmental heterogeneity within the field.
- Statistical Analysis: Variability was addressed through rigorous statistical methods, including ANOVA followed by Bonferroni correction, and data were presented as box plots to clearly show the distribution and interquartile range.
While some variability is inherent in field experiments due to uncontrollable factors (e.g., microclimatic differences, soil heterogeneity), we believe our approach effectively minimized its impact on the results. We have included a discussion (line 576 to 585, in red) of these measures in the revised manuscript to provide greater transparency. Thank you for highlighting this important aspect, and we hope this clarification addresses your concern.
Comment 6: It is recommended to add a mechanism discussion of the effect of Hariman treatment , especially the correlation in gene expression and virus accumulation, to enhance the logicality of the results. While the impact of Hariman on defense genes and plant hormones has been discussed, further exploration can be done on how these genes and hormones synergistically enhance the antiviral ability of plants. For example, the specific mechanisms of action of the SA and JA signaling pathways under Hariman treatment can be discussed. It is recommended to increase the discussion on the limitations of the research, such as the long-term stability of the handling effects by Hariman, the applicability under different environmental conditions, etc.
Response 6: We appreciate the reviewer’s suggestion to further explore the mechanisms behind the effect of Hariman treatment, particularly the correlation between gene expression and virus accumulation. As noted in the manuscript, Hariman significantly impacts plant defense-related genes and phytohormonal signaling pathways, enhancing plant resilience against CABMV infection. We agree that a deeper exploration of how these factors work synergistically to enhance the antiviral capacity of the plants will further strengthen the logical flow of our results. Regarding the specific signaling pathways, we have expanded the discussion on the roles of salicylic acid (SA) and jasmonic acid (JA) in the defense response under Hariman treatment. We now emphasize how SA, a key signaling molecule in plant defense, works in conjunction with JA to mediate a broader, more effective antiviral response. The balance and interaction between these two pathways could explain the observed reductions in viral accumulation and the enhanced defense response in treated plants.
Additionally, we would like to clarify that the current study included evaluations in both short- and long-term scenarios. In the short term, we assessed defense gene expression, phytohormones pathway, morpho agronomic parameters, and virus accumulation under greenhouse and field conditions. In the long term, we extended our evaluations to include defense gene expression, phytohormone-related genes, and productivity parameters such as fruit yield and quality. These last parameters were evaluated 10 months after treatment. These evaluations were conducted across three different locations: Seropédica and Rio de Janeiro (State of Rio de Janeiro), with a semi-humid tropical climate, and Brumado (State of Bahia) with a semi-arid climate. Furthermore, in another study, Santos-Jiménez et al. (2022b) evaluated the effect of Hariman in the municipality of Campos dos Goytacazes (State of Rio de Janeiro) that is 335 Km far from Seropédica. In all these studies, a positive effect of Hariman on productivity parameters and yields was consistently observed.
We have also added a section (line 543 - 575 in red) addressing the limitations of the study, including the need to further assess the long-term stability of Hariman’s effects over multiple growing seasons and its applicability across different environmental conditions, such as varying climate zones and soil types. While our study demonstrated Hariman’s effectiveness in both short- and long-term evaluations, future research will be crucial to understand its performance under diverse conditions and over extended periods. Such studies will provide a more comprehensive evaluation of Hariman’s potential for sustainable use in viral disease management.
REF:
Santos-Jiménez, J. L., de Barros Montebianco, C., Olivares, F. L., Canellas, L. P., Barreto-Bergter, E., Rosa, R. C. C., & Vaslin, M. F. S. (2022b). Passion fruit plants treated with biostimulants induce defense-related and phytohormone-associated genes. Plant Gene, 30, 100357.
Comment 7: Some terms are inconsistently used, such as "Hariman" sometimes written as ". It is recommended to unify it as "Hariman".
Response 7: Thank you for pointing this out. We unified the term Hariman in the hole MS. Where “Hariman®” was changed to “Hariman”, the word was highlighted in red.
Comment 8: Although the language of the article is generally fluent, there are still some minor errors, such as "phytohormones elicitation" should be changed to "phytohormone elicitation". It is recommended to conduct a comprehensive grammar and spelling check.
Response 8: Thank you for pointing that out. The MS was submitted to an English review by native English speaker's professional review.
